
1        **Spatiotemporial seismicity pattern of the Taiwan orogen**

Yi-Ying Wen[1, 2*], Chien-Chih Chen[3, 4], Strong Wen[1, 2], and Wei-Tsen Lu[1]
[1]Department of Earth and Environmental Sciences, National Chung Cheng University,
Chia-yi County 62102, Taiwan
[2]Environment and Disaster Monitoring Center, National Chung Cheng University,
Chia-yi County 62102, Taiwan
[3]Department of Earth Sciences, National Central University, Taoyuan City 32001,
Taiwan
[4]Earthquake-Disaster & Risk Evaluation and Management Center, National Central
University, Taoyuan City 32001, Taiwan
**Correspondence**: Yi-Ying Wen (yiyingwen@ccu.edu.tw)
**Abstract**
We investigate the temporal and spatial seismicity patterns prior to eight M>6 events
nucleating in different region of Taiwan through region-time-length algorithm and
analysis of self-organizing spinodal model. Our results reveal that the spatiotemporal
seismicity variations during the preparation process of impending earthquakes would
display distinctive pattern corresponding to the tectonic setting. Q-type events occur in
southern Taiwan and experience seismic quiescence stage prior to the mainshock.
Seismicity decrease of 2.5<M<4.5 events around the high b-value southern Central
Range, which contributes to accumulate the tectonic stress for preparing the occurrence
of the Q-type event. On the other hand, A-type events occur in central Taiwan and
experience seismic activation stage prior to the mainshock, which nucleated on the edge
of the seismic activation area. We should pay attention when accelerating seismicity of
3<M<5 events appears within the low b-value area, which would promote the
nucleation process of the A-type event.



## 1. Introduction


Seismic activity is related to spatiotemporal variation of stress field and state, and
seismicity changes prior to a large earthquake have been widely observed through
different techniques, e.g., b-values (Chan et al., 2012; Wyss and Stefansson, 2006),
PAST (Mignan and Giovambattista, 2008), PI (Rundle et al., 2003; Chen et al., 2005),
and RTL (Chen and Wu, 2006; Wen et al., 2016). Previous studies most focus on a
significant earthquake, therefore it is not easy to understand whether the properties of
seismic activation and quiescence pattern respond with the regional tectonic stress.
The Taiwan orogenic belt, which is an active and ongoing arc-continent collision
zone as a result of the Philippine Sea Plate (PSP) oblique colliding with the Eurasian
Plate (EP), is particularly complex due to the two adjacent subduction zones, the
Ryukyu trench and the Manila trench in the northeast and south of island, respectively
(Suppe, 1984; Yu et al., 1997). The frequent and significant seismic activities as well
as the rapid convergence rate of 85 to 90 mm/yr are well observed by the island-wide
GPS and seismic networks (Fig. 1). Suppe (1984) pointed out that the growth of Taiwan
orogenic belt shows propagation from north to south due to the oblique plate
convergence and the opposing subduction in the southern and northern parts of Taiwan.
In the southern Taiwan, the EP subducting eastward beneath the PSP is in the stage of
incipient arc-continent collision (Kao et al., 2000; Shyu et al., 2005). The coastal plain
and foothill region, which represent the southern tip of the fold-and-thrust belt in the
western Taiwan and show very low seismicity, mainly consist of the Miocene shallow
marine deposits and the Plio-Pleistocene foreland basin as well as mudstones. On the
other hand, the southern Central Range is mainly composed of Oligocence to Miocence
metamorphic slates and comprises ductile folds and cleavages as well as superimposed
faults. The central Taiwan, which is considered as the rapid to full collision, mainly



consists of Coastal Range, Central Range and Western Foothills (Shyu et al., 2005). A
myriad of active and thin-skinned structures are the products of the accretion of the
continental sliver to the continental margin. Over the last two decades, several moderate
earthquakes occurred in the various seismicity pattern and GPS velocity field regions.
This give us a chance to further investigate the spatiotemporal seismic pattern related
to the regional tectonic stress.

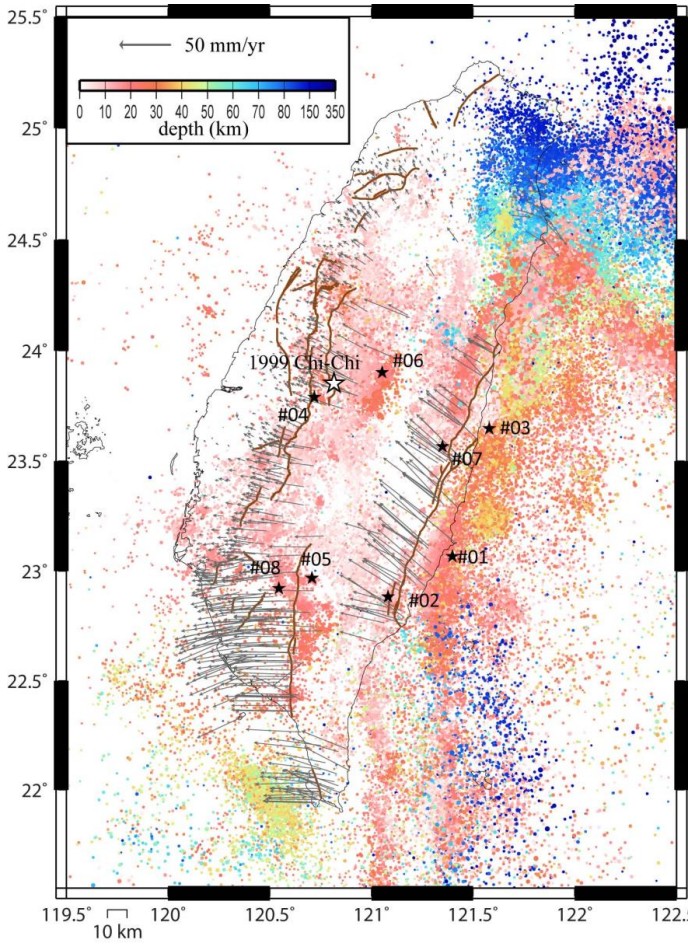

**Figure 1**: Horizontal velocities from 2002 to 2017 (Chen et al., 2018) and seismicity between 1991 to 2018. The white star shows the location of 1999 Chi-Chi earthquake, and the black stars represent the locations of the investigated events in this study. The active faults (thick lines) identified by the Central Geological Survey of Taiwan are also shown.


## 2. RTL Algorithm and Data

The region-time-length (RTL) algorithm (Sobolev and Tyupkin, 1997; 1999) is a statistical technique to detect the occurrence of seismic quiescence and activation by taking account of location, occurrence time and magnitude of earthquakes. The RTL value is defined as the product of the three dimensionless factors, $R$, $T$ and $L$:

$$R(x,y,z,t) = \left[ \sum_{i=1}^{n} \exp\left( -\frac{r_i}{r_0} \right) \right] - R_{bk}(x,y,z,t) \tag{1}$$

$$T(x,y,z,t) = \left[ \sum_{i=1}^{n} \exp\left( -\frac{t-t_i}{t_0} \right) \right] - T_{bk}(x,y,z,t) \tag{2}$$

$$L(x,y,z,t) = \left[ \sum_{i=1}^{n} \left( \frac{l_i}{r_i} \right) \right] - L_{bk}(x,y,z,t) \tag{3}$$

where $r_i$ is the distance between the investigated point $(x, y, z)$ and the $i$th prior event (with the occurrence time $t_i$ and rupture length $l_i$). $n$ is the number of prior events occurred in a defined space-time window with $r_i \leq 2r_0$ ($r_0$, characteristic distance) and $(t - t_i) \leq 2t_0$ ($t_0$, characteristic time-span). Rupture length $l_i$ is the function of earthquake magnitude ($M_i$), $\log l_i = 0.5M_i - 1.8$ (Kasahara, 1981). The weighted RTL value reflects the deviation from the background seismicity level ($R_{bk}$, $T_{bk}$ and $L_{bk}$) with negative value for seismic quiescence and positive value for an activation, respectively. Here, we adopt the $r_0 = 47.5$ km and $t_0 = 1.15$ yr model parameters based on previous studies for Taiwan (Chen and Wu, 2006; Wen et al., 2016; Lu, 2017; Wen and Chen, 2017).

For statistical analyses, the catalog completeness is the most important factor. Since 1991, the Taiwan Telemetered Seismographic Network (TTSN) (Wang, 1989) merged to the Central Weather Bureau (CWB) seismic network and updated to an integrated earthquake observation system, named Central Weather Bureau Seismic Network (CWBSN). Wang et al. (1994) pointed out that most of shallow earthquakes occurring in Taiwan distributed in the depth range less than 35 km. We used the



earthquake catalogue maintained by CWB for the entire Taiwan area with M≥2.5 and
depth≤35 km between 1991 to 2018 and applied a declustering procedure proposed by
Gardner and Knopoff (1974). Considering the sufficient background seismicity and
minimizing the influence of the 1999 Chi-Chi earthquake, we only selected the M>6
inland earthquakes between 2003 and 2016 in Taiwan. Since two events occurred in a
close space-time window would show high similarity in RTL function (Lu, 2017), we
neglected the event occurred within $2t_0$ and $r_0$ with respect to the last M>6 events. For
example, two M>6 events within a distance of 10 km struck the Nantou area on 27
March 2013 and 02 June 2013, we only analyzed the former event. Therefore, we have
eight qualified M>6 events, as list in Table 1.

**Table 1**: Earthquake parameters for the investigated events determined by CWB.

| No. | Date (UT) | Long. (deg.) | Lat. (deg.) | Depth (km) | $M_L$ |
|---|---|---|---|---|---|
| 1 | 2003/12/10 04:38:14 | 121.398 | 23.067 | 17.7 | 6.4 |
| 2 | 2006/04/01 10:02:20 | 121.081 | 22.884 | 7.2 | 6.2 |
| 3 | 2009/10/03 17:36:06 | 121.579 | 23.648 | 29.2 | 6.1 |
| 4 | 2009/11/05 09:32:58 | 120.719 | 23.789 | 24.1 | 6.2 |
| 5 | 2010/03/04 00:18:52 | 120.707 | 22.969 | 22.6 | 6.4 |
| 6 | 2013/03/27 02:03:20 | 121.053 | 23.902 | 19.4 | 6.1 |
| 7 | 2013/10/31 12:02:10 | 121.349 | 23.566 | 15.0 | 6.4 |
| 8 | 2016/02/05 19:57:26 | 120.544 | 22.922 | 14.6 | 6.6 |




**3.  Results**
**3.1 Temporal Seismicity variation**
The temporal variation in the RTL function represents the different stage of
seismicity rate change at the target location with respect to the background level. For
the consistency, we adopt 10-year catalogue as background for each investigated event.
Figure 2 shows the temporal variation of the RTL functions prior to the investigated
events. We can see that, before the occurrence of the investigated event, both seismicity
changes are observed: seismic quiescence stage for Nos. 1, 2, 5 and 8 (Q-type events
hereafter); seismic activation stage for Nos. 3, 4, 6 and 7 (A-type events hereafter),
respectively. Q-type events occurred on different locations in southern Taiwan, and
most, 3 among 4, of their temporal RTL functions reveal the seismic quiescence stages
during 2002-2004, which is before the occurrence of 2003 Chengkung earthquakes. The

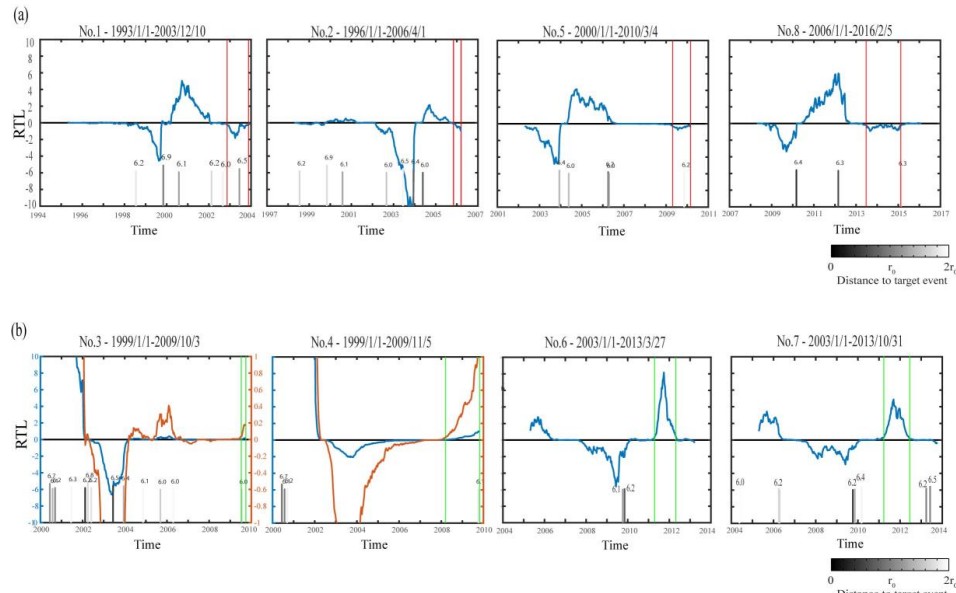

**Figure 2**: Temporal variation of the RTL function (blue line) for (a) Q-type events and (b) A-type events. The vertical red lines mark the seismic quiescence stage, and the vertical green mark the seismic activation stage. The bar chart represents the occurrence time of $M \geq 6.0$ events within the distance of $2r_0$ from the target event; each number above the bar is the magnitude.

seismicity increase (activation stage) took about two years following the 2003
Chengkung mainshock, i.e., event No. 1. We notice that the length of seismic
quiescence stage prior to the Q-type event might correspond to the magnitude. A-type
events all occurred in central Taiwan and located within $2r_0$ with respect to the 1999
Chi-Chi earthquake. Figure 3 shows the declustered seismicity distribution as a function
of time and latitude. It is noticed that the significant seismicity following the 1999 Chi-
Chi earthquake in the north of 23°N. Since the background seismicity of event Nos. 3
and 4 starting from 1999/01/01, the RTL functions are obviously affected by the
occurrence of the 1999 Chi-Chi earthquake. Therefore, we enlarge the vertical axis to

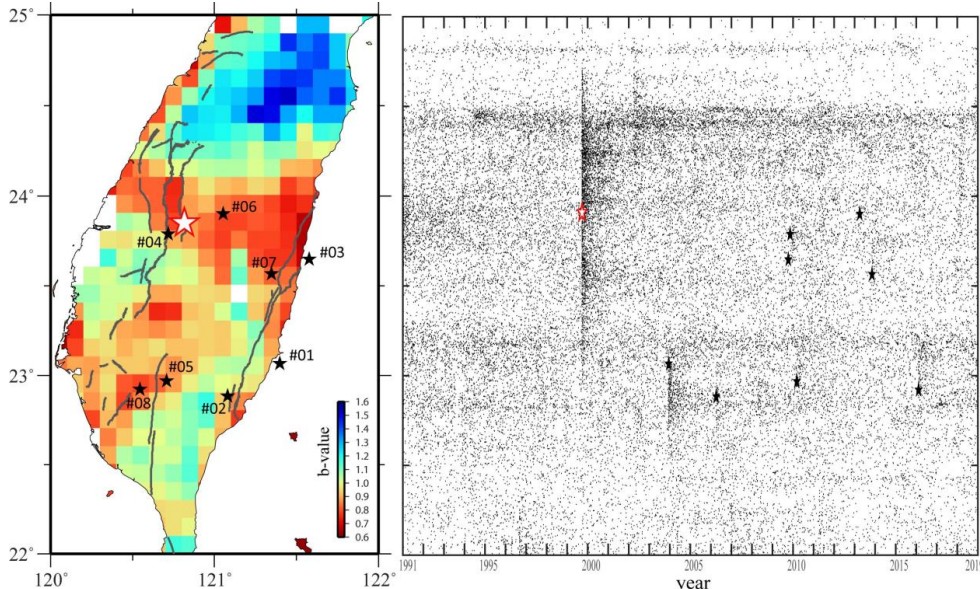

**Figure 3**: Map view of earthquake b-value and declustered seismicity distribution as a function of time and latitude. The white star indicates the 1999 Chi-Chi earthquake, and the black stars represent the investigated events in this study. The active faults (thick lines) identified by the Central Geological Survey of Taiwan are also shown.


accentuate the seismicity variation prior to event Nos. 3 and 4. As shown in Fig. 2, the
temporal RTL functions of A-type events most show seismic activation stage between


2004-2006, which correspond to the seismicity increase following the 2003 Chengkung
mainshock. However, for the A-type event, we could not see the relationship between
the length of seismic activation stage and the magnitude.

**3.2 Spatial Seismic Activation/Quiescence Distribution**
Since Q-type and A-type events located in southern and central Taiwan,
respectively, it would be worth to examine the spatial pattern of their abnormal
seismicity stages. Wen and Chen (2017) pointed out that various seismic activation or
quiescence processes of about 2-4 years were found prior to some events occurred in
Taiwan. Thus, for the consistency, we select the last abnormal stage within four years
prior to the investigated events, as marked by red vertical lines for quiescence stage of
Q-type events and green vertical lines for activation stage of A-type events,
respectively. Then, we calculate the summation of selected period to generate the
seismic quiescence/activation distribution. Considering the definition of the weighted
RTL function, the sufficient number of background seismicity should be regarded as a
criterion. Using the declustered catalogue during 1991 to 2016, we set up two
conditions for each grid to strengthen the reliability: (i) the total number of events
within the grid area of 0.1°×0.1° must be more than 26 (i.e., at least 1 event occurred
every year on average); (ii) the total events within a circle of $2r_0$ in radius must be more
than 9360 (i.e., at least 30 events occurred every month on average). Similar with
previous studies (e.g., Huang et al., 2001; Huang and Ding, 2012), Figure 4 shows that
Q-type events occurred on the edge of the seismic quiescence area and A-type events
occurred on the edge of the seismic activation area.



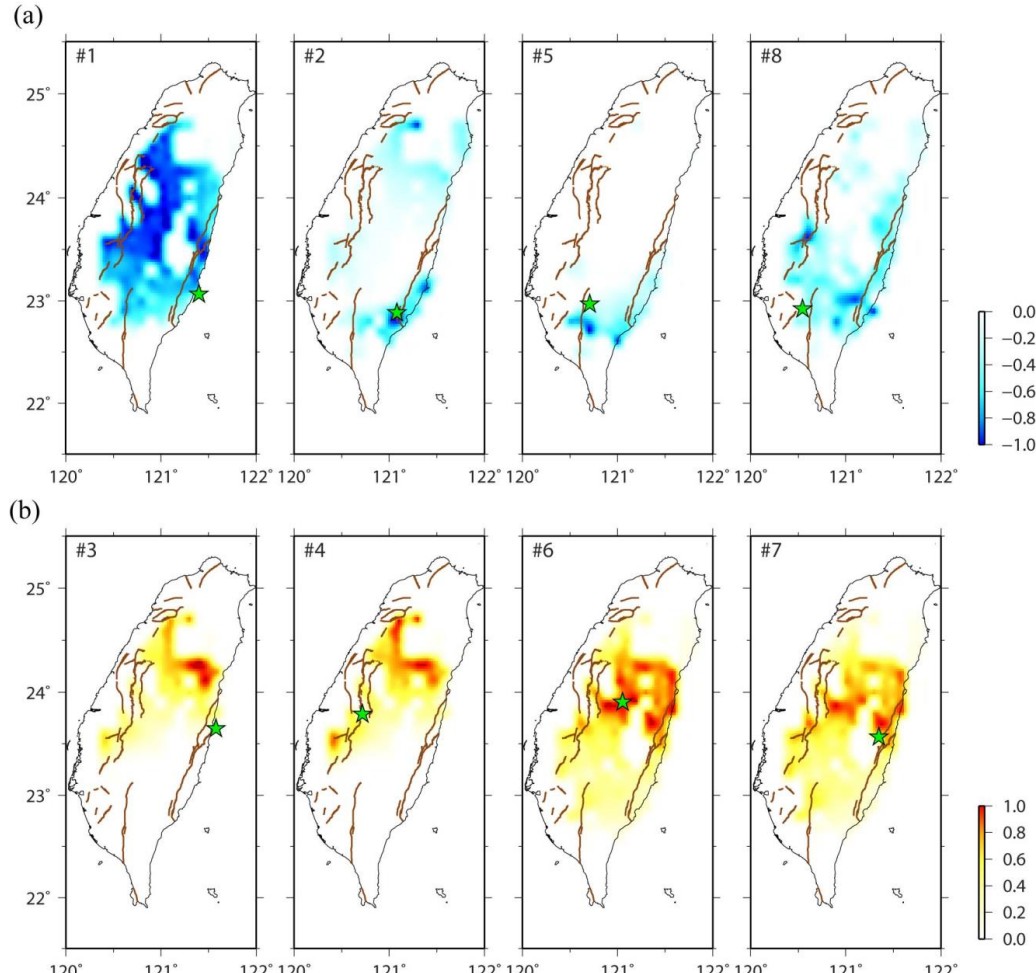

**Figure 4**: (a) The summed seismic quiescence map for the selected time window of the temporal RTL function of Q-type events, and (b) the summed seismic activation map for the selected time window of the temporal RTL function of A-type events. Stars represent the locations of the investigated events. The active faults (thick lines) identified by the Central Geological Survey of Taiwan are also shown.


**4. Discussion**
**4.1 Spatiotemporal Characteristics of Seismicity Changes**
The RTL analysis accounts for the background seismicity prior to the investigated
event. For example, the analysis for event No. 1 (i.e., 2003 Chengkung earthquake)



used the declustered catalogue between 1993/01/01 to 2003/12/09 as background
seismicity for each grid. It is noticed that four A-type events occurred in two different
years: event Nos. 3 and 4 in 2009, as well as event Nos. 6 and 7 in 2013 (Fig. 2).
Therefore, the RTL analyses account for almost the same background length for event
Nos. 3 and 4 as well as for event Nos. 6 and 7, respectively. As the temporal RTL
functions show the seismic activation stage prior to the mainshocks during the similar
period, we could expect the similar seismic activation maps, as shown in Fig. 4.
Furthermore, the seismic quiescence stage of event No. 5 occurred in the similar period
of the seismic activation stage of event No. 3 (Fig. 2), and the seismic quiescence area
of event No. 5 plays as a complement to seismic activation area of event No. 3 (Fig. 4).
On the contrary, although event Nos. 3 and 7 occurred on the close locations, the
difference of 4-year background seismicity affects the weighting of the deviation. For
example, as shown in Fig. 2, the seismic quiescence stage during 2007-2009 revealed
in the temporal RTL function of event No. 7 is evaluated as the background seismicity
level in the temporal RTL function with respect to event No. 3. On the other hand, Wen
and Chen (2017) pointed out that an abnormal seismic stage revealed with various
background period cannot be produced by chance. The temporal RTL functions of five
events (Nos. 1-5 in Fig. 2) accounting for different background periods all reveal the
seismic quiescence stage before the occurrence of event No. 1. This phenomenon is
consistent with the seismic quiescence map of event No. 1 (Fig. 4) and Z-value map of
Wu et al. (2008) that, the seismicity activity decreased during 2002-2003 for large area
in Taiwan. In addition, the widespread seismic activation distribution of Nos. 6 and 7
(Fig. 4) also respond to the seismicity activity increasing during 2011-2012 (Nos. 6-8
in Fig. 2). Overall, the seismic quiescence and activation maps show some
characteristics: (i) the seismicity decrease is revealed in the southern Central Range




prior to the Q-type mainshocks; (ii) it seems the boundaries around 23.2°N and 24.5°N
for the abnormal seismicity distributions, which coincide with the distribution of
declustered seismicity in Fig. 3.

Rundle et al. (2000) proposed the self-organizing spinodal (SOS) model for

characteristic earthquakes and suggested that small earthquakes occurred uniformly at
all times while occurrence rate of the intermediate-sized earthquakes varied during the
earthquake cycle. Chen (2003) investigated the SOS behavior of the 1999 Chi-Chi
earthquake and revealed the seismic activation of moderate-size (5<M<6) events prior
to the mainshock. Here, we also calculate the cumulative frequency-magnitude
distributions for these eight events, using the same catalog periods of the RTL analysis.
For each investigated event, we only compared the distribution diagrams of long-term
(background period) and abnormal seismic stage marked in Fig. 2, within a radius of
25 km with respect to the epicenter. As shown in Fig. 5, cumulative frequency-
magnitude distributions of long-term seismicity (red dots) generally exhibit linear
power law distributions. For the Q-type events, the cumulative frequency distributions
of seismic quiescence stage (black dots) appear lack in number of 2.5<M<4.5 events
(Fig. 5a), and the lacking level corresponds to the seismic quiescence distribution near
the epicenter (Fig. 4). This indicates that, within the seismic quiescence stage before
the occurrence of Q-type event, the quiescence of 2.5<M<4.5 activity contributes to the
accumulation of tectonic stress. On the other hand, the cumulative frequency
distributions of seismic activation stage of the A-type events (black dots in Fig. 5b)
display that the seismic activation of 3<M<5 events within the seismic activation stage
before the occurrence of A-type earthquake can be found, and this is similar with the
result of the 1999 Chi-Chi earthquake (Chen, 2003). Event Nos. 6 and 7, which locate




very close to the high seismic activation area (Fig. 4), display obvious increase in
number of 4<M<5 events during the seismic activation stage (Fig. 5b).

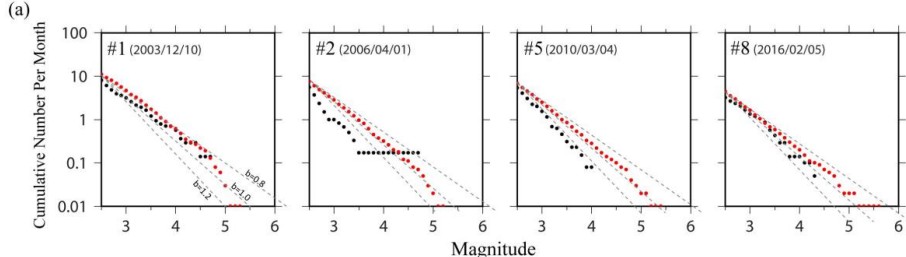

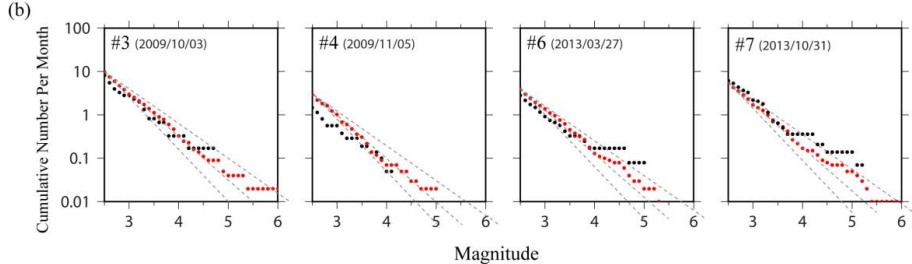

**Figure 5**: The cumulative frequency-magnitude distributions prior to the investigated
events. Red and black dots represent the long-term and abnormal seismic stage
marked in Fig. 2, respectively.


Event No. 4 occurs only one month later than event No. 3, however, the seismic
activation stage of event No. 4 is much longer than that of event No. 3. Furthermore,
the cumulative frequency distributions of seismic activation stage of event No. 4 display
the lower intercept, which represents the overall decreasing seismicity within this
seismic activation stage (Fig. 5b). Here, we further divide the seismic activation stage
of event No. 4 into three periods for discussion: (i) P1: 2008/02-2009/03 before the
seismic activation stage of event No. 3; (ii) P2: 2009/04-2009/09 matching the seismic
activation stage of event No. 4; and (iii) P3: 2009/10 between the occurrences of event
Nos. 3 and 4. The seismic activation distributions in Fig. 6 are all normalized with
respect to the maximum RTL value of seismic activation distribution of event No. 4.



We can see that, before the seismic activation stage of event No. 3 during 2008/02-
2009/03 (P1), the location of event No. 3 indeed shows no seismic activation, as
revealing in the temporal RTL function (Fig. 2b). On the other hand, for the location of
event No. 4, the seismic activation remains through all three periods P1-P3. Combining
with the overall decreasing seismicity indicated by the lower intercept in Fig. 5(b), it
suggests that this seismic activation prior to event No. 4 is mainly contributed by the
relatively accelerating activity of 3.5<M<4 events.

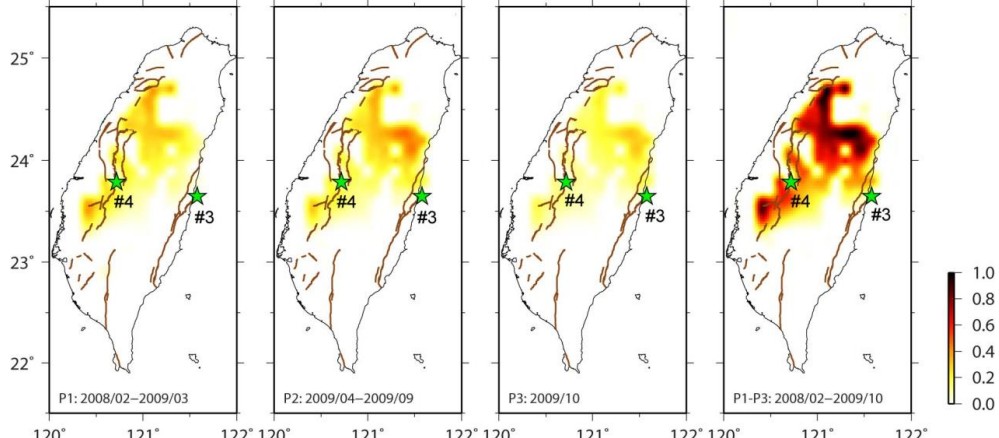

**Figure 6**: The summed seismic activation map for different period of the seismic
activation stage prior to event No. 4. Stars represent the locations of the event
Nos. 3 and 4. The active faults (thick lines) identified by the Central Geological
Survey of Taiwan are also shown.


**4.2 Implication of Tectonic Setting**
Several major active faults in southern Taiwan were identified, and most of them
were dominated by thrust movement. Some strike-slip structures, e.g. the Zuochen and
Hsinhua fault, played as the transfer structures between those thrust faults (Ching et al.,
2011; Deffontaines et al., 1994, 1997; Rau et al., 2012). These transfer structures
develop around 23°N, where is the northern limit of the Wadati-Benioff zone (Kao et
al., 2000) and close to the seismicity boundary indicated in Figs. 2 and 4. Geodetic data


revealed the various rate and orientation horizontal shortening with rapid uplift rates in
the southern Taiwan (Fig. 1), which might be caused by the underplating beneath the
Central Range sustain crustal thickening and exhumation (Simoes et al., 2007). The
seismicity in southern Central Range is active but showing significant heterogeneity in
faulting types (Chen et al., 2017), and high b-values suggest the predominance of small
earthquakes in this region (Fig. 2 and red dots in Fig. 5a). Wen et al. (2016) revealed
the seismicity decrease in the southern Central Range prior to the 2010 Jiashian
earthquake (i.e., event No. 5). The seismicity rate change can be considered as a proxy
for the stress state change (Dieterich, 1994; Dieterich et al., 2000), and both variations
in Coulomb stress and seismicity rate play important roles contributing to the nucleation
process of impending earthquakes (Wen et al., 2016). Since this high b-value region in
southern Central Range is observed seismicity decrease (2.5<M<4.5 events) before the
occurrence of Q-type events, it can be an indicator of stress change.

Many devastating earthquakes with surface rupture occurred in this region,

including the 1935 M 7.1 Hsinchu-Taichung earthquake, the 1951 Longitudinal Valley
earthquake sequence and the 1999 Chi-Chi earthquake (Lee et al., 2007; Chen et al.,
2008; Lin et al., 2013). Hsu et al. (2009) derived the consistent orientations of principal
strain-rate and crust stress axes in central Taiwan, which implies that faulting style
corresponds to stress buildup accumulating from the interseismic loading. They also
pointed out that, for the central Taiwan, the small events tend to surround the locked
fault zone, where the major earthquakes might occur, during the interseismic period.
The 1999 Chi-Chi earthquake is such case ruptured the area near the end of décollement
with a high contraction rate (Dominguez et al., 2003; Hsu et al., 2003; 2009). In addition,
similar with the 1999 Chi-Chi earthquake, the A-type events occurred on the low b-
value area surrounded by small and active events. Chen and Wu (2006) derived the



temporal RTL function of the 1999 Chi-Chi earthquake showing similar pattern to the
A-type events with the activation stage prior to the mainshock. Furthermore, Wu (2006)
calculated the seismic activation map of 1999 Chi-Chi event, and the author also found
that the 1999 Chi-Chi mainshock occurred on the edge of the seismic activation area,
where is a low b-value region. This is similar with the seismic activation maps of A-
type events, which display the hot-spot pattern contracting within the low b-value area
(Fig.s 2 and 4). The nucleation of A-type mainshock can be attributed to the
perturbation of background seismicity (3<M<5 events) by the stress state change
(Dieterich, 1994; Dieterich et al., 2000).
The cumulative frequency distributions of long-term seismicity in Fig. 5 reveal the
b-value of 0.8-1.0 around these eight events, which consist with the pattern shown in
Fig. 3. However, the cumulative frequency distributions of long-term seismicity exhibit
different trends of magnitude larger than 4.5 for two type events that the seismicity for
M>4.5 events is lower in area around Q-type event but higher in area around A-type
event. Event Nos. 1, 2, 3 and 7, occurred in eastern Taiwan with averaged GPS velocity
of about 60 mm/yr (Fig. 1), and the cumulative frequency distributions of long-term
seismicity display the high intercept (Fig. 5). This rapid convergence rate roughly
remains in the west part of southern Taiwan, which indicates that only few shortening
is consumed through east to west in the southern Taiwan. This corresponds to the active
seismicity of small earthquakes as revealed by the high intercept of the cumulative
frequency distributions of long-term seismicity for event Nos. 1, 2, 5 and 8 (Fig. 5).
Therefore, the quiescence of 2.5<M<4.5 activity contributes to accumulate the tectonic
stress for preparing the occurrence of the Q-type event. On the other hand, the
shortening rate is obviously consumed in the mountain area of central Taiwan.
Therefore, the lowest intercept of the cumulative frequency distributions of long-term


seismicity for event No. 4 (Fig. 5) reflects the slow GPS velocity and low seismicity in
the west part of the central Taiwan (Fig. 1). Tectonic stress accumulating from the
interseismic loading with the perturbation of the accelerating activity of 3<M<5 events
would promote the nucleation process of the A-type event.

**5. Conclusion**

Through the statistical analyses of recent large earthquakes occurred in Taiwan, we

summary various temporal and spatial seismicity patterns prior to the earthquakes
nucleating in different region of Taiwan:

• Q-type events occurred in southern Taiwan, with the northern boundary of

23.2°N, and experienced seismic quiescence stage prior to the mainshock.

Seismicity decrease of 2.5<M<4.5 events in the high b-value southern

Central Range could be an indicator of stress change related to the

preparation process of such type event.

• A-type events occurred in central Taiwan and experienced seismic

activation stage prior to the mainshock, which nucleated on the edge of the

seismic activation area. We should pay attention when accelerating

seismicity of 3<M<5 events appears within the low b-value area.

Our results reveal that the spatiotemporal seismicity variations during the

preparation process of impending earthquakes would display distinctive pattern
corresponding to the tectonic setting. However, it is not so clear about the mechanisms
causing these different phenomena, further study is still needed.



**Data Availability :** The seismic data is available in the Geophysical Database
Management System (GDMS, https://gdms.cwb.gov.tw/). A Chinese manual for data
access from the GDMS is on the website.

**Author contributions**: Conceptualisation, YYW, CCC; Investigation, YYW, WTL;
Validation, Formal analysis, Writing - original draft preparation, YYW; Writing -
review & editing, YYW, CCC, SW.

**Competing interests**: The authors declare no conflicts of interest.

**Acknowledgments:** We thank Central Weather Bureau (CWB) of Taiwan for
providing seismic data. This research was supported by the Ministry of Science and
Technology in Taiwan with grant: MOST 110-2116-M-194-018. The Taiwan
Earthquake Center (TEC) contribution number for this article is ****.




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
