# Peer review of "Yi-Ying Wen1, 2\*, Chien-Chih Chen3, 4, Strong Wen1, 2, and Wei-Tsen Lu1"

_Natural Hazards and Earth System Sciences, 2022_

## Author Comment (AC1)

Dear reviewer,

Thank you very much for reviewing our paper titled " Spatiotemporial seismicity pattern of the Taiwan orogen". We have read the review carefully and have accordingly made substantive modifications to the manuscript and explained the details in the response letter below. The manuscript was revised to address all changes marked in red.

Sincerely,
Yi-Ying Wen and co-authors

**Reviewer #1:**

**Comment:** The conclusion of this paper was drawn with only one RTL parameter. We know that even a small change in the RTL parameters can change the conclusion, which is dangerous. See Nagao et al (2011) for a comparison of multiple RTL parameters.

**Reply**: We understand reviewer's concern. Therefore, we follow the systematical procedure of correlation analysis over pairs of RTL results proposed by Huang and Ding (2012) to obtain the optimal model parameters, $\widetilde{r}_0$ and $\widetilde{t}_0$, of each event to diminish the ambiguity in determining the characteristic parameters. We calculate various combination of $r_0$ (ranging between 25 and 80 km with a step of 2.5 km) and $t_0$ (ranging between 0.25 and 2.0 yr with a step of 0.05 yr). After testing many criterion sets, the criterion coefficient $C_0$ = 0.6 and criterion ratio $W_0$ =0.5 are acceptable for each event, which means at least 50% of the total combination pairs with correlation coefficient $C \geq C_0$ = 0.6. Since we do not attempt to catch the seismic precursor but focus on the seismicity changes related to the regional tectonics, which might become useful hint for potential seismic-hazard assessment. Here, we obtained the average $\widetilde{r}_0$= 49.6 km and average $\widetilde{t}_0$ = 1.16 yr. These model parameters are similar to those of previous studies for Taiwan (Chen and Wu, 2006; Wen et al., 2016; Lu, 2017; Wen and Chen, 2017). We have added the description in L. 85-99 and Appendix.

**Comment:** Both Sobolev et al. and Wyss reported that seismic quiescence occurs shortly before the impending earthquake (e.g., Wyss et al., 2004., Huang et al, 2001, Huang and Nagao, 2002).

**Reply**: Several studies also find seismic quiescence occurring prior to the impending earthquake in Taiwan, e.g. the 2003 Chengkung earthquake (event No. 1) (Wu et al., 2008), the 2010 Jiashian earthquake (event No. 5) (Wen et al., 2016), and the 2016 Meinong earthquake (event No. 8) (Wen and Chen, 2017). On the other hand, Chen and Wu (2006) shows the seismic activation before the occurrence the 1999 Chi-Chi earthquake. In addition, Huang (2019) also reveals the seismic activation before the occurrence the 2008 Ms7.3 Yutian, China earthquake. It suggests that either seismic quiescence or activation stage can occur prior to the impending earthquake.

**Comment:** Furthermore, it cannot be proved that the seismic quiescence that occurred many years ago is not related to the earthquake that occurred several years later, but it is meaningless for practical earthquake prediction.

**Reply**: Various seismic activation or quiescence processes of about 2-4 years were found prior to some events occurred in Taiwan (Chen and Wu, 2006; Wen et al., 2016; Wu et al., 2008) and worldwide (Huang et al., 2002; Huang and Ding, 2012). Thus we only consider the last abnormal stage within four years prior to the investigated events. In this study, we do not attempt to catch the seismic precursor or apply earthquake prediction but want to investigate the characteristics of spatial and temporial seismicity pattern related to the regional tectonics, which might become useful hint for potential seismic-hazard assessment. We have made out goal clearer in L. 63-69.

**Comment:** The RTL value is the product of the standard deviations. It takes a value of -8 when it is -2 sigma in terms of time, -2 sigma in space, and -2 sigma in terms of the size of the earthquake. Sobolev et al., also find that the RTL value fluctuates and basically makes sense for the seismic quiescence that exceeds around -8.

Nagao et al (2011) proposed another algorithm for L in the RTL method (RTM algorithm). This is because L appears twice in the definition of L proposed by Sobolev et al. It seemed this dual appearance of Ri seems to be in contradiction to the original concept of the RTL algorithm.

Therefore, the reviewer has previously contacted Dr. Sobolev directly and asked, "Why does the distance (Ri) appear twice in L's formula?" Dr. Sobolev's answer was, "This is to make it easier to detect seismic activation."

This means that once a relatively large earthquake occurs in the vicinity of the RTL calculating point, the RTL value becomes large and discontinuous (e.g. Nagao et al., 2011).

**Reply**: Considering reviewer's suggestion, we also apply the RTM algorithm proposed by Nagao et al. (2011) under 72 parameter sets for each investigated event. Figure R1 shows some temporal variation of RTM functions of event No. 1 obtained with various parameter sets. The RTM functions indeed show some differences with various parameter sets, however, they generally exhibit similar pattern with the seismic quience stages between 1999 to 2000 and prior to event No. 1 as well as the seismic activation stage between 2000 to 2002. This pattern is consistent with our RTL function of event No. 1. Here, we further adopt the same characteristic parameter set of our RTL model to calculate the RTM function of each event, and both functions display very similar trend with minor differences, as shown in Figure R2. The reason could be that, for these eight events, there are no large earthquakes occurred in the vicinity of the epicenter. The bar chart in Figure R2, which represents the occurrence time of M ≥ 6.0 events within the distance of $2r_0$ from the target event, also support this explanation. We also add this comparison in the Appendix.

Through visual inspection, we found that most investigated events would not show the quiescence with value exceeding -8 before the target event. This criterion was neither applied in previous studies (Chen and Wu, 2006; Huang et al., 2001; 2002; Huang and Ding, 2012; Wu et al., 2008). We love to share reviewer's idea of 2 sigma. Statistically speaking, 2 sigma is about related to the 95% confindence, which we consider a overly optimistic and high standard regarding the noisy nature of earthquake catalogues. Also, as reply above, we do not attempt to catch the seismic precursor or apply earthquake prediction but want to investigate the characteristics of spatial and temporial seismicity pattern related to the regional tectonics. Since our RTL results are obtained with

characteristic parameter set from the systematical procedure of correlation analysis. We keep using RTL algorithm in this study.

[Figure]

Figure R1: Temporal variation of the RTM functions of event No. 1 obtained with various parameter sets. The bar chart represents the occurrence time of M ≥ 6.0 events within the distance of rmax from the target event; each number above the bar is the magnitude.

[Figure]

Figure R2: Temporal variation of the RTL (solid line) and RTM (dotted line) functions for (a) A-type events and (b) B-type events. The bar chart represents the occurrence time of M ≥ 6.0 events within the distance of 2r0 from the target event; each number above the bar is the magnitude.

**Comment:**Furthermore, although it is written in Japanese, there is a paper that seismic quiescence and activation occur in pairs (Matsumura, 2005). To briefly summarize Matsumura's hypothesis, "Seismic quiescence is recognized as a macro-scale view due to stress reallocation caused by activation of local seismic activity, and for the micro-scale, there seems a locally activated region does exist."

**Reply**: Thank you for sharing this hypothesis, and we also agree with it. Wen et al. (2016) also suggests that the activation period following the 2010 Jiashian earthquake could be related to the increased Coulomb stress change (ΔCFS).

**Comment:** In the current content, it is hard to say that this paper properly uses the characteristics of the RTL algorithm, and it cannot be said that it explains the characteristics of Taiwan's seismic activity very much.

In conclusion, a major extensive revision is required, and it is judged that publication is not possible at the present stage.

**Reply**: Deeply thank you for the thoughtful review of this paper and the comments to help us improve this paper. The manuscript has now been revised carefully based on all the comments.

**Comment:** Minor Problems:

Figure 2, q-type: Much greater quiescence than the authors point out as "quiet" has appeared before that. Furthermore, the quiescence value (RTL value) is extremely small, and it seems like a range of fluctuations.

a-type: Similar to the q-type, a large activation period may have appeared before that, or it may take more than a year and a half from the end of activation to the actual occurrence of an earthquake. The orange curve is drawn in the two graphs, but there is no explanation for it.

**Reply**: This is a very interesting question. Should it be the biggest order of the quiescence or activation prior to the impending earthquake? In my thought, this is similar to the situation that most studies show the target event occurred on the edge of the seismic quiescence area. The seismicity rate change corresponds to the stress change, and the occurrence of mainshock can be interpreted as a perturbation of background seismicity by the stress state change (Dieterich, 1994; Dieterich et al., 2000). Wen et al. (2016) found that the 2010 Jiashian mainshock was occurred on a region with stress state changing from decrease to increase. It indicates that the large earthquake could occur on the region with anomalous seismicity and stress state change. Again, we should emphasize that we intend to figure out the possible relationship between the seismicity change pattern and the regional tectonics, therefore, we focus on the characteristic of the seismic variation stage prior to the target event. As discussion in the manuscript, although some RTL values are small, they can still represent the meaningful seismic stage. For example, the seismicity increase following the 2003 Chengkung mainshock (event No. 1) can be identified by the temporal RTL functions of some close events with seismic activation stage between 2004-2006, including event Nos. 2, 5, 6, 7 and 3.

Thank you for the careful review. We have added the explanation of the orange curve in the caption.

**Comment:** LINE 180:

Rundle et al., 2000 does not exist in the references. According to the authors, this paper describes the SOS model. There is no description of the self-organizing spinodal model in Rundle et al 2003.

**Reply**: We have corrected it.

---

## Author Comment (AC2)

Dear reviewer,

Thank you very much for reviewing our paper titled " Spatiotemporial seismicity pattern of the Taiwan orogen". We have read the review carefully and have accordingly made substantive modifications to the manuscript and explained the details in the response letter below. The manuscript was revised to address all changes marked in red.

Sincerely,
Yi-Ying Wen and co-authors

**Reviewer #2:**
**Comment:** In the paper by Wen et al., Spatiotemporial seismicity pattern of the Taiwan orogen, the RTL algorithm is applied to the seismicity of Taiwan to investigate the seismicity patterns prior to M>6 events. Based on this analysis, the authors recognize two types of events, the ones that experience seismic quiescence before the mainshock (Q-type) and the ones that show seismic activation prior to the mainshock (A-type). Although the results seem interesting, there are some major issues with the analysis, which are discussed in the following. Therefore, I recommend major revisions before the paper can be reconsidered for publication.
**Reply**: We deeply thank reviewer for the thoughtful review of this paper. The comments below that allowed us to greatly improve this paper. The manuscript has now been carefully revised based on all comments.

**Comment:** 1) Revise the Introduction section and discuss the main objectives of the paper and how these will be accomplished.
**Reply**: Following reviewer's suggestion, we have added the description in L. 63-69.

**Comment:** 2) The RTL algorithm is based on characteristic parameters, such as the characteristic distance and time. The authors adopt these parameters based on previous studies in Taiwan. However, it should be shown and discussed how sensitive are the results of the RTL algorithm on these parameters.
**Reply**: Considering both reviews' suggestions, as reply for previous comment, we follow the systematical procedure of correlation analysis over pairs of RTL results proposed by Huang and Ding (2012) to obtain the optimal model parameters, $\widetilde{r_0}$ and $\widetilde{t_0}$, of each event to diminish the ambiguity in determining the characteristic parameters. We calculate 828 parameter sets of $r_0$ (ranging between 25 and 80 km with a step of 2.5 km) and $t_0$ (ranging between 0.25 and 2.0 yr with a step of 0.05 yr). After testing many criterion sets, the criterion coefficient $C_0 = 0.6$ and criterion ratio $W_0 = 0.5$ are acceptable for each event, which means at least 50% of the total combination pairs with correlation coefficient $C \geq C_0 = 0.6$. Here, we obtained the average $\widetilde{r_0} = 49.6$ km and average $\widetilde{t_0} = 1.16$ yr. These model parameters are similar to those of previous studies for Taiwan (Chen and Wu, 2006; Wen et al., 2016; Lu, 2017; Wen and Chen, 2017). This supports the feasibility of our characteristic parameters. We have added the description in L. 85-99 and Appendix.

**Comment:** 3) The authors discuss that a complete catalogue is a significant factor for the RTL analysis and use the events with M≥2.5. Is this the magnitude of completeness since 1991 for Taiwan? Please justify.

**Reply**: Wu and Chiao (2006) pointed out that the CWBSN greatly enchanced the earthquake monitoring capability in Taiwan and reduced Mc to a value of about $M_L$=2.0 since the end of 1993, which is consistent with another work (Huang, 2020), as shown in Figure R3. For the RTL analysis, only event No. 1 involves 1-yr data of 1993, and it does not affect the result much.

[Figure]

Figure R3: The annual variation of Mc in Taiwan. (Huang, 2020)

**Comment:** 4) The results of the RTL analysis, presented in Fig.2, further show negative RTL values and seismic quiescence stages prior to the quiescence stage identified and marked by the authors. How can these stages affect future large events and the main conclusions of the paper?

**Reply**: As reply in previous comment, this temporal phenomenon is similar to the situation of spatial pattern that most studies show the target event occurred on the edge of the seismic quiescence area. The seismicity rate change corresponds to the stress change, and the occurrence of mainshock can be interpreted as a perturbation of background seismicity by the stress state change (Dieterich, 1994; Dieterich et al., 2000). Wen et al. (2016) found that the 2010 Jiashian mainshock was occurred on a region with stress state changing from decrease to increase. It indicates that the large earthquake could occur on the region with anomalous seismicity and stress state change. Again, we emphasize that we intend to figure out the possible relationship between the seismicity change pattern and the regional tectonics, therefore, we focus on the characteristic of the seismic variation stage prior to the target event. As discussion in the manuscript, although some RTL values are small, they can still represent the meaningful seismic stage. For example, the seismicity increase following the 2003 Chengkung mainshock (event No. 1) can be identified by the temporal RTL functions of some close events with seismic activation stage between 2004-2006, including event Nos. 2, 5, 6, 7 and 3.

**Comment:** 5) The resolution of Fig.2 should be improved.
**Reply**: We indeed generate high-quality figures, however, the converted PDF file shows lower resolution.

**Comment:** 6) Discuss how the spatial variations of the b-value, shown in Fig.3, were calculated.
**Reply**: We have added the details in L. 257-265.

**Comment:** 7) In Fig.3, spatiotemporal clustering of seismicity is still visible following large events, although the catalogue is declustered. Are the aftershocks effectively being removed?
**Reply**: As shown in Figure R4, the cumulative number of earthquakes from declustered catalog suggests the aftershocks being removed effectively.

[Figure]

Figure R4: (a) The cumulative number of earthquakes from the original CWB catalog (red line) and the declustered catalog (black line). (b) The declustered seismicity distribution as a function of time and latitude.

**Comment:** 8) Line 133. How the four years time span prior to the investigated events was selected?
**Reply**: Various seismic activation or quiescence processes of about 2-4 years were found prior to some events occurred in Taiwan (Chen and Wu, 2006; Wen et al., 2016; Wu et al., 2008) and worldwide (Huang et al., 2002; Huang and Ding, 2012). Thus we only consider the last abnormal stage within four years prior to the investigated events.

**Comment:** 9) Lines 139-143. How the criteria i) and ii) were selected? Are the results sensitive to these criteria?

**Reply**: Since the influence weight of the RTL function is contributed from the location, occurrence time and magnitude of the prior events, Wen and Chen (2017) suggests that the sufficient number of background seismicity should be considered as a criterion. They set up the criteria through many testing, and we apply the same criteria here.

**Comment:** 10) In Fig. 4, explain what the colorbars represent. Similarly for Fig.6.
**Reply**: Thank reviewer's reminder, we have added more explaination in L. 168-169, L. 238 and the figure captions.

**Comment:** 11) Overall, a better justification of the presented results is required.
**Reply**: In this study, we use two different methods to investigate the characteristics of seismicity behavior for eight earthquakes. We do not intend to group them spatially in the beginning, but the results do. This gives another point of view for the seismicity pattern in different tectonics. Our results, which show many consistencies with several previously studies, are reliable and meaningful.

**Comment:** Some minor comments to the text concern:
1) A few issues with English language throughout the text should be improved.
**Reply**: We have sent our manuscript to a professional English Language Editing company before submitting this revision. Please find the Editing Certificate in the attachment.

**Comment:** 2) Spatiotemporal rather than Spatiotemporial.
**Reply**: We have corrected it.

**Comment:** 3) In Page 2, Lines 32-34, refer to the full names of these methods before using the abbreviations. Also add a brief discussion to introduce them properly.
**Reply**: We have added the full names of these methods in L. 34-37 and introduce three of them in various parts of the manuscript.

**Comment:** 4) Add Rundle et al. (2000) to the list of references.
**Reply**: We have added it.

[Figure]

**Editing Certificate**

This document certifies that the manuscript

**Spatiotemporal seismicity pattern of the Taiwan orogen**

prepared by the authors

**Yi-Ying Wen, Chien-Chih Chen, Strong Wen, and Wei-Tsen Lu**

was edited for proper English language, grammar, punctuation, spelling, and overall style by one or more of the highly qualified native English speaking editors at AJE.

This certificate was issued on **June 2, 2022** and may be verified on the AJE website using the verification code **E01A-4EC6-8190-DB96-7D06** .

[Figure]

Neither the research content nor the authors' intentions were altered in any way during the editing process. Documents receiving this certification should be English-ready for publication; however, the author has the ability to accept or reject our suggestions and changes. To verify the final AJE edited version, please visit our verification page at aje.com/certificate. If you have any questions or concerns about this edited document, please contact AJE at support@aje.com.

AJE provides a range of editing, translation, and manuscript services for researchers and publishers around the world. For more information about our company, services, and partner discounts, please visit aje.com.